# An Extensive Literature Review on Underwater Image Colour Correction

**DOI:** 10.3390/s21175690

**Published:** 2021-08-24

**Authors:** Marinos Vlachos, Dimitrios Skarlatos

**Affiliations:** Department of Civil Engineering and Geomatics, Cyprus University of Technology, Saripolou Street 2-6, Limassol 3036, Cyprus; dimitrios.skarlatos@cut.ac.cy

**Keywords:** UW, colour correction, image restoration, artificial intelligence

## Abstract

The topic of underwater (UW) image colour correction and restoration has gained significant scientific interest in the last couple of decades. There are a vast number of disciplines, from marine biology to archaeology, that can and need to utilise the true information of the UW environment. Based on that, a significant number of scientists have contributed to the topic of UW image colour correction and restoration. In this paper, we try to make an unbiased and extensive review of some of the most significant contributions from the last 15 years. After considering the optical properties of water, as well as light propagation and haze that is caused by it, the focus is on the different methods that exist in the literature. The criteria for which most of them were designed, as well as the quality evaluation used to measure their effectiveness, are underlined.

## 1. Introduction

The synthesis of an underwater (UW) image is an intricate procedure that is affected by a plethora of effects that are usually ignored in images captured in air (e.g., uneven spatial illumination, colour dependent attenuation, backscatter, etc.) [1]. For that reason, many researchers are engaged with the topic of UW image processing and ways on how to improve the overall quality of their work through the years, either from a geometric or visualisation standpoint. Based on the above statement, the scope of colour correction in UW images is a topic that has received considerable attention in the last decades.

An extensive review was conducted back in 2010 by Raimondo Schettini and Silvia Corchs [2], where they evaluated and discussed the state-of-the-art. The authors divided image processing into two different points of view; image enhancement and image restoration. According to the authors, the goal of image enhancement is to generate a visually appealing image without relying on any physical models. Image restoration techniques, on the other hand, utilise the image formation model to restore the degraded image. Such models utilise the optical properties of water, camera to object distance and camera-specific information, such as the response function. Even though methodologies such as those presented in [3,4,5,6,7,8,9,10] and thoroughly described by Schettini and Corchs, are considered outdated, they were the foundations for today’s advancements in UW image colour correction and restoration. Since then, more methods have risen as well as new avenues regarding UW image processing, such as various artificial intelligence algorithms.

### 1.1. Optical Properties of Water

As described by Wang et al., UW images always show a green-bluish colour cast, which is driven by different red, green and blue light attenuation ratios [11]. The water properties that control light attenuation of water and thus the scene appearance is dependent on scattering and absorption. Attenuation coefficients control how the light decays exponentially as a function of the distance that it travels [12]. Pure waters are optically clear mediums with no suspended particles; only the interaction of light with molecules and ions causes light to be absorbed in pure water. [13]. Short visible wavelengths, such as red, are absorbed first, followed by green and then blue. As a result, just 1% of the light reaching the water’s surface reaches a depth of 100 m. [14]. 

The absorption and scattering coefficients of water have been a hot topic for many years. Jerlov classified waters into three separate oceanic kinds and five unique coastal kinds in 1951 [15]. Following Jerlov’s work many years later, methods such as the one described in [16,17] aim to determine the inherent optical properties of Jerlov’s water types. 

Instead of a colour correction technique in the RGB space, a mathematical model for spectral analysis of water characteristics was proposed in [18]. On the same note, the optical categorisation of natural water bodies was utilised by Akkaynak et al. [17] to determine the positions of all physically important RGB attenuation coefficients for UW imaging. The range of wideband attenuation coefficients in the ocean is restricted, and the authors demonstrated that the usual transition from wavelength-dependent attenuation β(λ) to wideband attenuation β(c) is not as simple as it appears contradicting the image formation model.

### 1.2. Image Haze

The goal of clear images captured UW is very significant in ocean engineering [19,20]. Other than the evaluation and determination of the physical properties of water and the impact they have on the degradation of the colours of a scene, capturing UW images is even more challenging due to haze. As Chiang and Chen describe in [21], “haze is caused by suspended particles such as sand, minerals, and plankton that exist in lakes, oceans, and rivers. As light reflected from objects propagates toward the camera, a portion of the light meets these suspended particles”. Various techniques of removing the haze effect from UW images and thus compensating for light scattering distortion were introduced in the literature [7,10,22].

### 1.3. Recent Advancements and the Work of This Paper

The goal of this paper is to present and discuss modern contributions in the domain of UW image colour enhancement and restoration. We will try to give an overview of many significant contributions, discuss certain results and give our unbiased opinion regarding the impact that the state of the art has on improving the way we observe the UW environment. The selection of methodologies discussed in the paper was based on relevancy to the topic, as well as the final visual results each method provides. Additionally, the selection of each contribution was based on when it was published. The methods that will be presented are mostly published from 2015 to 2020 since previous review papers focus on methods published prior to that time frame. In total, 80 papers were screened related to the topic of UW colour correction. Following that, a separation of classes was made where three major classes were identified, separating the methods to image enhancement, image restoration and artificial intelligence (AI). For image enhancement, 9 papers were selected, the same as image restoration, where for AI, 13 papers were selected for review. The search of the evaluated papers was performed mostly through the Google Scholar search engine. Some of the search queries that were used are the following:Underwater Image Colour CorrectionUnderwater Image Colour RestorationUnderwater Image Colour EnhancementUnderwater Image Colour Correction using AIImage Colour Restoration with Neural NetworksImage Colour Correction using water physical propertiesImage Colour Correction in deer watersColour Correction with ROV/AUVImage Formation Model and Colour Correction

The structure of the paper is as follows: Section 2 will give some overview of the various contributions, while Section 3 will discuss the pros and cons of each category, as well as thoughts, problems and applicability for each one. Finally, in Section 4, conclusions and suggestions are presented.

## 2. Various Contributions in Recent Years

This section will discuss in brief various UW image colour correction methods that were contributed to throughout recent years. This section is divided into three subsections. The subsections are dedicated to image enhancement methods, image restoration methods and finally, artificial intelligence methods. 

### 2.1. Image Enhancement Methods

As it is described by [2], to create a more visually appealing image, image enhancement employs qualitative, subjective criteria rather than relying on a physical water characteristics model. These methods are often easier and faster than deconvolution techniques.

Ancuti et al. [23] suggested a simple fusion-based approach for enhancing UW photos using a single input by successfully blending multiple well-known filters. As the authors state, their method proved successful in improving UW footage of dynamic situations. Additionally, this study created a robust white balancing technique tailored for UW sceneries that was also validated after an exhaustive evaluation. The method described is one of the first that managed to demonstrate the practicality of an UW enhancement methodology for many non-trivial applications, such as segmentation, image feature matching and dehazing. Although the approach produces acceptable results when dealing with images of low depth, it is limited with images of deep scenes obtained with weak strobe and artificial light. Even if some augmentation is possible in such circumstances, the blue hue persists. Furthermore, insufficient lighting makes it difficult to consistently retrieve the scene’s far reaches. The restriction of the suggested method is exacerbated even more with hardware and polarisation-based approaches, which in general perform better in such instances due to the additional information provided.

A first proposal for colour correction of UW images by using the lαβ colour space is presented in [24]. To increase image contrast, chromatic components’ distributions are white balanced, and histogram cut-off and stretching of the luminance component are done. Under the assumption of a grey world and homogeneous lighting of the scene, the experimental findings show that this strategy is successful. For close-range acquisition in a downward direction, such as seabed mapping or recording artefacts in the foreground (UW photography), or less complicated situations with less light change, these assumptions are acceptable. The suggested solution takes a single image and does not require any filters or prior scene information. The scene is considered to be consistently lighted by a light source. This indicates that the light intensity is mostly steady across the image. In addition, the authors translated their grey world hypothesis into lαβ space and tested it on two UW images taken in distinct places. They proved that this approach is acceptable for removing unwanted colour casts in UW imaging. Based on the experimental results, greenish-blue components in the images are fully eliminated, even in grey tones. White balancing of components α and β is used to adjust colour, while the luminance component l is handled to increase the contrast. In comparison to simply altering the distributions of colour channels around a grey point as in RGB space, correcting the colour casts with regard to the white point in lαβ space appears to be more realistic. The authors argue that the grey-world assumption in the lαβ space is turned into a “white-world” assumption as a result of this transformation. This contribution has been and still is influential in future researchers, as is indicated later in this paper. Figure 1 shows a comparison between various correction methods.

Even though this method is a strong tool for the 3D reconstruction and visualisation of UW cultural heritage, it significantly depends on the presence of red in the scene. As is shown in Figure 1, when the scene does not contain enough red, the corrected object seems to look grey, as is shown in the last row in column 2 of Figure 1. Therefore, for the method to be successfully applied, either the site must be in shallow waters, meaning that natural lighting will be present, or there needs to be enough artificial lighting through the use of strobes. 

Based on colour correction and a non-local prior, Wu et al. [25] suggested a specific process for UW image restoration. To render and fix foggy UW images, they first remove the colour distortion, followed by dehazing to eliminate the backscattering effects. This work’s contributions are: (1) the introduction of a global background light estimation algorithm based on quadtree subdivision and UW imaging optical properties; (2) the presentation of an UW colour enhancement method using depth compensation that accounts for different rates of colour degradation; and (3) it applies a non-local prior to acquire a transmission map. As it is explained by the authors, the dark channel of an underwater image will have higher intensity in regions farther from the camera; the scene depth can be qualitatively reflected by the dark channel. The authors propose a depth compensation method, in which a multi-channel guided image filter (MGIF) is used to modify the depth image before colour correcting it. Despite the fact that this technique makes use of the scene’s geometry, the outcomes are comparable to those of the other approaches covered in this section. 

For UW photos, a colour correcting approach based on local surface reflectance statistics is described in [26]. This scheme decomposes the original colour distorted image into several non-overlapped patches. For each patch, the illumination is estimated based on statistics. Following the above, the true reflectance for each image block is obtained. The method seems to provide some good results, with the most reliable being when the number of patches used is big enough, as the colour of the scenery is smoothly restored, whereas on the other hand, when the number of patches is small, the colour contrast of red is very intense and creates an artificial and not realistic result. It needs to be mentioned that the datasets used by the authors are downloaded from the internet, and based on the examples shown in their work, the algorithm was only tested in images taken in shallow waters with the presence of natural light.

In [27], the authors’ colour correcting approach is based on the use of the grey-world assumption in the Ruderman-lαβ opponent colour space, considering the chrominance variation across the scene. The authors proposed a local estimation of the illuminant colour by average computation with a moving window around all pixels. By doing this, the correction to the nonuniform illumination is adapted. In addition, the authors perform contrast enhancement and colour saturation, using a perceptually uniform logarithmic stretching that ensures a natural and plausible colour appearance. Bianco’s and Neumann’s contributions have provided chromatically pleasing results. The results of the global approach they used are especially encouraging since they can provide realistic results by moving the overall average of α and β channels, while a cut-off of 1% of both luminance histogram limits and stretching is used to improve the image contrast.

In Figure 2 the results from three real cases are presented. The results are very promising since the colour of the scenery is restored, providing realistic results. It should be mentioned here that this method also works well in shallow water due to the presence of natural light and thus red colour. 

Some other contributions on the topic of UW image enhancement methods exist in the literature and are proposed by Nurtandio Andono et al. [28], Ancuti et al. [29], Zhao et al. [30] and Peng et al. [31]. These contributions provide good results on image enhancement using techniques such as the dark channel prior or the image blurriness to compensate for the lack of scene depth. 

As a common theme of this subsection, all the methods that were showcased work relatively well in shallow waters, but none of them were tested in deep waters where the presence of red is obsolete. Additionally, most of these methods show limitations in their performance on objects that are far away from the camera.

### 2.2. Image Restoration Methods

As described in [2], image restoration is an inverse problem based on image formation models that try to restore a deteriorated image using a model of the degradation and the original image formation. These approaches are rigorous, but they need a large number of model parameters (such as attenuation and diffusion coefficients that describe water turbidity), which sometimes are available in tables and might vary a lot, affecting the final restored images. Another important required parameter is the camera to object distance, which is also referred to as depth in many cases.

Bryson et al. [32] propose an automated method for rectifying colour discrepancy in UW photos gathered from diverse angles while building 3D structure-from-motion models. The purpose of this research is to image large scale biological environments, which prohibits the use of colour charts due to the sensitivity of marine ecosystems to seabed disturbances. The authors deploy a “grey-world” colour distribution to focus on colour constancy. This means that surface reflectance has a grey-scale distribution that is independent of scene geometry. The underwater images were collected using a stereo camera setup on an AUV, with two 1360-by-1024-pixel cameras; the left camera was a colour-Bayer sensor and the right camera a monochrome sensor. The authors explain and test image transformation algorithms, such as grey world transformation:(1)$$I_{y}\left(u,v,\lambda \right)=m\left(\lambda \right){Ι}_{x}\left(u,v,\lambda \right)+n\left(\lambda \right)$$
where ${Ι}_{x}\left(u,v,\lambda \right)$, in given image *x*, is the initial intensity of a specific pixel (*u*, *v*) for band *λ* in image *x*
$I_{y}\left(u,v,\lambda \right)$ is the intensity in image *y*, where *m*(*λ*) and *n*(*λ*) are scaling, and offset constants. The mean $\mu_{y}\left(\lambda \right)$ and variance $\sigma_{y}^{2}\left(\lambda \right)$ of the pixel histogram are essentially a function of the original transformation parameters and image statistics:(2)$$\mu_{y}\left(\lambda \right)=m\left(\lambda \right)\mu_{\chi }\left(\lambda \right)+n\left(\lambda \right)$$
(3)$$\sigma_{y}^{2}\left(\lambda \right)=m{\left(\lambda \right)}^{2}\sigma_{\chi }^{2}\left(\lambda \right)$$
where $\mu_{\chi }\left(\lambda \right)$ and $\sigma_{x}^{2}\left(\lambda \right)$ are the mean and variance of a band in the original image. These equations may then be utilised to create a linear transformation that produces an image band with whatever mean and variance are needed:(4)$$m\left(\lambda \right)=\sqrt{\frac{\sigma_{y}^{2}}{\sigma_{\chi }^{2}\left(\lambda \right)}}$$
(5)$$n\left(\lambda \right)=\mu_{y}-m\left(\lambda \right)\;\mu_{\chi }\left(\lambda \right)$$
where $\mu_{y}$ and $\sigma_{y}^{2}$ are the required variance and mean. In the end, the author proposes an image transformation accounting for range dependent attenuation for a given dataset of UW images. The equations, in this case, are very similar to the grey-world transformation that is showcased above:(6)$$I_{y}\left(u,v,\lambda \right)=m\left(u,v,\lambda ,d\right){Ι}_{x}\left(u,v,\lambda \right)+n\left(u,v,\lambda ,d\right)$$
(7)$$m\left(u,v,\lambda ,d\right)=\sqrt{\frac{\sigma_{y}^{2}}{\sigma_{\chi }^{2}\left(u,v,\lambda ,d\right)}}$$
(8)$$n\left(u,v,\lambda ,d\right)=\mu_{y}-m\left(u,v,\lambda ,d\right)\;\mu_{\chi }\left(\lambda u,v,\lambda ,d\right)$$
where $\mu_{\chi }\left(\lambda u,v,\lambda ,d\right)$ and $\sigma_{x}^{2}\left(\lambda u,v,\lambda ,d\right)$ for a depth *d* and a band *λ* are the mean and variance of all intensities in a given batch of N images at the pixel position (*u*, *v*) for depth *d* and band *λ*. The authors suggest that a potential method for estimating $\mu_{\chi }\left(\lambda u,v,\lambda ,d\right)$ and $\sigma_{x}^{2}\left(\lambda u,v,\lambda ,d\right)$ is to divide pixel intensities into given *d* (distance from the camera) bins and obtain the mean and variance for each bin. Unfortunately, the approach necessitates a large number of samples per bin, resulting in under-sampling of particular ranges and pixel positions. To solve this, the authors used a different method, creating a scatter plot of image intensity vs. range for each pixel position (Figure 3), measuring one point for every image of the dataset. If $\mu_{R}$ is the surface’s predicted mean reflectance, then the average image intensity acquired from pixels (*u*, *v*), band *λ*, and range *d* is:(9)$$\mu_{x}\left(u,v,\lambda ,d\right)=a\left(u,v,\lambda \right)\mu_{R}e^{-b\left(u,v,\lambda \right)d}+c\left(u,v,\lambda \right)$$
where a(u,v,λ), b(u,v,λ) and c(u,v,λ) are pixel location and image band parameters. The authors proceed to estimate the terms $a\left(u,v,\lambda \right)\mu_{R}$ (as one), b(u,v,λ) and c(u,v,λ) via a non-linear least-square fit using Levenberg–Marquardt optimisation [33]. 

The authors show through Figure 3 that the image intensity decays exponentially the further an object is from the camera. Below, in Figure 4, the results of this colour restoration method are shown and compared with standard non-depth-based correction techniques.

Based on the results shown above, this method seems effective since the image contrast and intensities match the overall scene, and at the same time, the close regions of the images are darker thanks to the restoration. One of the downsides of this method is the large number of images and depth maps required.

Another method proposed in [34] by Galdran et al. is the automatic red-channel UW image restoration. The study is carried out in two main parts. First, the pixels that reside at the maximum scene depth with respect to the camera are chosen, assuming that the deterioration is distance-dependent; this position naturally corresponds to the maximum values in the original image’s red channel. In the case of UW scenarios, the one that has the lowest red component obtains the best results in their experiments. Here the authors, although they mention the need of depth information of the scene, they do not describe any workflows they applied to obtain it. The first step of the process aims at the estimation of water light from the red channel. The following step is to estimate the transmission from the red channel with a straightforward extension of the dark channel prior method [35]. The colour correction phase is carried out in two parts. First, the authors deal with the vectorial transmission, where they distinguish three separate transmission maps and three components of the water light, one for each colour. After this step, the attenuation coefficients are estimated, and this procedure is described as Weighted WaterLight. Galdran et al. tested their algorithm on various images aiming to recover the contrast and visibility of the scenery. This method again seems to work very well for close-range objects but loses its effectiveness on objects that are further from the camera. Additionally, based on the images that are used, it needs the presence of natural lighting and specifically red.

In 2016, an interesting and detailed methodology was introduced in [36]. Here the authors proposed a formation model for calculating the true colour of scenery taken from an UW automated vehicle equipped with a stereo camera setup with strobes:(10)$$I_{\lambda }=k\left[C\left(a\right)a\left(\lambda \right)\overset{i=1}{\underset{Nl}{\sum }}\left({Ρ}_{\Phi \iota }cos\theta_{i}e^{-b\left(\lambda \right)\left(r_{c}+r_{li}\right)}\right)+B\left(\lambda \right)\right]$$
where:(11)$$C\left(a\right)=1+C\alpha 2\alpha^{2}+C\alpha 4\alpha^{4}+C\alpha 6\alpha^{6}$$

Cα2, Cα4, Cα6 are polynomial coefficients and C(α) being a vignetting coefficient. ${Ρ}_{\Phi \iota }$ is the light source power, which can be set as ${Ρ}_{0\iota }$ = 1 as the authors suggest. $\theta_{i}$ is the angle that the light hits the object, *r_c_* and *r_li_* are the distances of the object from camera and light source. a(λ) is the albedo/reflectance of the object. b(λ) is the attenuation coefficient. B(λ) is the backscattered signal, which can be modelled as:(12)$$\left(\lambda \right)=\frac{\beta \left(\lambda \right)}{b\left(\lambda \right)}\left[1-e^{-b\left(\lambda \right)r_{c}}\right]$$

*β*(*λ*) is the backscattering coefficient. 

The exposure constant is known from the recorded shutter speed of the camera, and it is believed to be factor *k*, which is not clearly specified. Using several co-registered measurements of each point from distinct image views, the authors approach concurrently calculates the reflectance of the scene points as well as the camera/water parameters. This approach computes a maximum-likelihood estimate for the unknown parameters (*b*, *β*, *C**α*2, *C**α*4, *C**α*6, *a*1, *a*2, …, *a*N)T on the 3D terrain surface created by structure-from-motion using non-linear least-square and Levenberg–Marquardt optimisation. After computing the camera and water parameters, the reflectance of any point on the seafloor surface may be calculated using the following formula:(13)$$a\left(\lambda \right)=\left[\frac{I\left(\lambda \right)}{k}-B\left(\lambda \right)\right]{\left[C\left(a\right)\overset{i=1}{\underset{Nl}{\sum }}\left({Ρ}_{\Phi \iota }cos\theta_{i}e^{-b\left(\lambda \right)\left(r_{c}+r_{li}\right)}\right)\right]}^{-1}$$

The mean image intensities (*I*) for the reprojected triangle in each image of each triangle make up the observation vector *z* for a particular channel.
(14)$$z=\left[I_{1}^{1},\;I_{1}^{2},\ldots ,I_{1}^{M1},I_{2}^{1},\;I_{2}^{2},\ldots ,I_{2}^{M2},\;I_{N}^{1},\;I_{N}^{2},\ldots ,I_{N}^{MN}\right]$$

Only the image intensities of a small number of surface points are used to estimate these parameters. The authors evaluate their results with the use of calibrated colour charts. Since there are no ground truth data, to evaluate the effectiveness of this method, the RGB values of calibrated colour charts that are present in the corrected scene are compared with the RGB values of the same charts in the air. The results of this process are shown in Figure 5 below.

The authors in [36] proposed a methodology that manages to restore the colour of UW images as they are captured without the presence of water. Although the method is very effective, it also requires a very specific and expensive setup with an autonomous UW vehicle (AUV) equipped with a fully calibrated camera and strobe setup. Unfortunately, this methodology is difficult to use for general cases of UW imagery where divers capture the images due to the lack of info regarding the positions of the strobes with respect to the camera.

Akkaynak and Treibitz in [37] analyse the current UW image formation model (Equation (15)) and derive the physically valid space of backscatter using oceanographic measurements as well as images acquired from NikonD810 and a Sony RX100 cameras, demonstrating that the wideband coefficients of backscatter differ from those of direct transmission, despite the fact that the current model portrays them as the same. As a result, the authors suggest a new UW image generation equation that takes these variations into account and validates it using in situ UW experiments (Equation (16)).
(15)$$I_{c}=J_{c}\cdot e^{-\beta_{c}z}+B_{c}^{\infty }\cdot \left(1-e^{-\beta_{c}z}\right)$$
where: $I_{c}:the\;UW\;scenery$$J_{c}:the\;unattenuated\;scenery$$\beta_{c}:wideband\;attenuation\;coefficient$z:distance from the camera$B_{c}^{\infty }:\mathrm{wideband}\;\mathrm{veiling}\;\mathrm{light}$

(16)$$I_{c}=J_{c}\cdot e^{-\beta_{c}^{D}\left(vD\right)z}+B_{c}^{\infty }\cdot \left(1-e^{-\beta_{c}^{B}\left(vB\right)z}\right)$$
where *vD* and *vB* the vector containing the coefficient dependencies *vD* = {*z*, *ρ*, *E*, *Sc*, *β*} and *vB* = {*E*, *Sc*, *b*, *β*}.



$\beta_{c}^{D}:\mathrm{direct}\;\mathrm{transmission}\;\mathrm{attenuation}\;\mathrm{coefficient}\;$

$\beta_{c}^{B}:\mathrm{backscatter}\;\mathrm{attenuation}\;\mathrm{coefficient}$

E:irradiance

$S_{c}:sensor\;response$

ρ:reflectance of the object



The same authors implemented their work in [38], creating a pipeline for colour reconstruction. While the updated model is physically more accurate, it contains more parameters that make it challenging to use. The authors introduce the Sea-thru methodology in this paper, which explains how to estimate these parameters for improved scene recovery. Sea-thru is a key step toward allowing sophisticated computer vision and machine learning algorithms to access vast UW datasets. The datasets used for these experiments were captured using a Sony 7R Mk III with a 16–35 mm lens and a Nikon D810 with a 35 mm lens. This work has achieved excellent and perhaps the most promising results in recent years. Provided it is described well enough to be replicated in a future paper or made open access, it will contribute to several UW applications. The coefficient related to backscatter varies with sensor, ambient lighting and water type, according to these two studies. Additionally, it is different from the coefficient associated with the direct signal. The authors used their own dataset of raw images with their corresponding depth maps in order to test five different scenarios; S1: Simple contrast stretch, S2: Former model with an incorrect estimate of Bc, S3: Former model, with a correct estimate of Bc and $\beta_{c}^{D}=\beta_{c}^{B}=\beta_{c}$, S4: Revised model, with a correct estimate of Bc, and Jc, S5: Sea-thru implementation. Again, the authors here evaluated the effectiveness of the model with the use of colour charts. As the authors conclude, Sea-thru is a key step in allowing strong computer vision and machine learning algorithms to access vast UW datasets, which will aid in the domain of UW research. One downside of this method though, is the need for natural light in the scene. That means that this methodology, as viable as it is, can only be used on datasets captured above 20 m UW, making it obsolete for datasets where the presence of natural light does not exist. The results produced by each scenario are shown in Figure 6.

The contributions from [37] have been utilised in [39]. In this study, the authors proposed a system that calculates the attenuation coefficients required to solve the imaging formation model equation, allowing the colour correction of images onboard an UW robot. Here a colour chart of known calibrated values is utilised to estimate the direct and backscattering attenuation coefficients and then allow for the estimation of the unattenuated scene using the revised image formation model. 

Bekerman et al. [12] have developed a method for measuring the attenuation parameters and veiling light directly from the images. The authors then use a typical image dehazing approach to retrieve the whole physical model of the scene after the parameters have been obtained. The transmission map, depth map, veiling light and undistorted scene are all part of the physical model. This method was constructed based on the knowledge that “the scattering of light in the medium between the object and the camera adds an additive component to the image, meaning that the scattering increases as the distance between the object and the camera grows”. Veiling light is the saturation value of this, and it happens when there are no objects in the line of sight (LOS). The veiling light value is commonly computed from visible sections in the image that include no objects and is considered to be constant across the scene. Because this assumption is not robust, the authors have contributed to the aforementioned method.

A study regarding image restoration methods was conducted by Berman et al. [40], where the authors, due to the lack of ground truth data for UW sceneries, took multiple stereo images that contain colour charts. These charts were then used in stereo pair images to obtain the true distance from the camera. 

A very recent contribution regarding UW image restoration was published in [41]. Here the authors proposed a novel algorithm based on the complete UW image formation model. Their work consists of the estimation of the transmission with the observation that the scene distance is inversely proportional to the geodesic colour distance from the background light. 

### 2.3. Artificial Intelligence Methods

The domain of machine and deep learning (ML and DL) has bloomed the last decade and has contributed a lot in the field of marine sciences and thus the UW environment. Through this impact, many tools and algorithms were developed for the purpose of UW image restoration. 

A first approach to AI implementation was introduced many years ago with the introduction of stochastic processes, such as Markov Random Field (MRF). The problem of colour restoration using statistical priors is addressed by [42] using an energy minimisation approach; this is used for the colour recovery of UW images. The concept is founded on the assumption that an image may be modified as a sample function of a stochastic process, i.e., as an MRF. They see colour correction as a problem of utilising the training image patches to give a colour value to each pixel of the input image that best describes its surrounding structure. The authors use UW web images as “ground truth” or acquisition of UW video by an aquatic robot. The “ground truth” images that the authors use are images downloaded from the internet that represent various UW sceneries with realistic colours. The results of [42] are divided into two scenarios. Scenario 1 uses “ground truth” images, depletes them, and finally applies the algorithm. Scenario 2 uses “ground truth”, images with artificial light from the robot and trains the algorithm to correct depleted images accordingly. In this case, the depleted images are taken without the presence of light at frame t+δt, where the “ground truth” images are taken at frame t. For Scenario 1, the authors produced results using different training sets. The “ground truth” datasets for this work were the afformentioned internet images. In scenario 2, even though the method depends on frames with light as ground truth data, the algorithm produces visually good results.

Based on the results presented by the authors, the methodology works very well in both scenarios. Unfortunately, some issues arise regarding reproducing the results of the study in other cases. For Scenario 1, the algorithm is trained on a specific dataset, meaning that the results produced are based on that. This means that if the algorithm is applied to images captured under different environments, the results might heavily deviate from the truth. Additionally, for Scenario 2, the need for such specific equipment is discouraging for most cases.

In [43], the authors developed a spatial chromatic-MRF model that only considers the spatial domain of images and is chromatic since the model’s variables only consider the scene’s chromatic information, ignoring the luminance channel L in the CIELab colour space. Their model is based on [42]. This work presents two different results. One is Visual Attention White World Assumption (VAWWA), in which they estimate illuminant based on a relevant mask constructed from points of interest found using the method described in [44], and one with MRF-Bayesian Belief Propagation (BP) with Automatic Colour Adaptive Training.

The use of MRF is also utilised in [45] to restore the colour of UW video. A multiple colour space analysis and processing phase is performed to automatically recover the colour in an image frame of a video sequence that will be used as a training set in the MRF model. Based on current colour restoration approaches documented in the literature, the authors suggest an automated methodology to build a viable training set for an MRF aimed to recover the colour in a video sequence. The whole strategy is adaptive in nature, with the goal of creating a dynamic training set that reacts to changes in UW sceneries. It is assumed that the histograms of the scenes in a video series may be used to detect whether there is a substantial change in the chromatic channels α and β of the lαβ colour space (of the colour deprived frames). This incident may then be used as an indicator by the algorithm to adjust the training set and proceed with the restoration of the following batch of frames with comparable chromaticity. The base of building the training set is Adapted White-World Assumption with Contrast Limited Adaptive Histogram Equalisation (CLAHE) and illuminant adjustment. At every substantial change of scene, the training set creation procedure begins by applying a gamma correction on the RGB colour deficient frame and converting it to the lαβ colour space. This has a result of colour shifting that is corrected. Based on the results presented in [43] and [45], the WWA method that is proposed produces satisfying results, but when the conditions are challenging, the final result suffers, as is stated by the authors. Visual Attention White World Assumption (VAWWA) eliminates ambient light in an image quite well, however, in some circumstances, the remotion tends to eliminate other colours that are not part of the lighting, resulting in a greyish image.

In [46], the authors created a multiterm loss function that includes adversarial loss, cycle consistency loss and SSIM (Structural Similarity Index Measure) loss, which was inspired by Cycle-Consistent Adversarial Networks. They introduce a new weakly supervised model for UW image colour correction that translates the colour from UW scenes to air scenes without the need for explicit pair labelling. The model directly outputs a coloured image as if it was taken without the water. The results that the authors present are somewhat visually pleasing and also can improve the performance of feature image matching algorithms. Additionally, a user study was conducted in order to evaluate the performance of the method since there is no measure specifically established for UW image colour correction.

The authors in [47] created the UW Denoising Autoencoder (UDAE) model, a deep learning network built on a single denoising autoencoder employing U-Net as a Convolutional Neural Network (CNN) architecture, as a contribution to restoring the colour of UW images. The suggested network considers accuracy and computation cost for real-time implementation on UW visual tasks utilising an end-to-end autoencoder network, resulting in improved UW photography and video content. The authors propose a UDAE network that is specialized in UW colour restoration, and they suggest that they achieve faster processing than the state of the art methods. They create a synthetic dataset using a generative deep learning method. The dataset has a combination of different UW scenarios. The results provided can improve the visual aesthetic of a scene and give relatively good results in real UW scenes, even though the training of the algorithm was done using synthesized datasets. In Figure 7, the results of this method compared with the results from UW Generative Adversarial Network (UGAN) [48] are shown.

As it is supported by the authors, in many cases UDAE can provide as good or even better results than UGAN. This method can preserve the details of the images providing a realistic result, but it is not effective in all real scenarios since the training is done by utilising synthesised UW images and not real sceneries.

Other Generative Adversarial Networks (GAN) based studies that are considered as important contributions are the CycleGAN [49], MyCycleGAN [50], WaterGAN [51], UWGAN [52] and UGAN [48], as well as the work proposed by Yang et al. [53], Liu et al. [54] and Gulrajani et al. [55].

## 3. Discussion

All the methods that were discussed in Section 2 have contributed significantly over the years. Specifically, each of the three categories have extensively been researched. Each category has its pros and cons, best and worst case of implementation, problems and limitations. In this section, we discuss and give our thoughts for all three categories in an unbiased manner. Table 1 gives a short description, characteristics and datasets that certain methods were applied on, as well as what kind of quality assessment was conducted by each of the authors listed.

Image enhancement methods, although they are usually simpler and faster and can produce pleasing results, they do not rely on any physical water properties or any formation model. That means that in many cases, the criteria for the enhancement are subjective. These factors lead to different behaviours according to the scene. That can lead to certain limitations when we apply a specific method to non-favourable scenes with varying conditions as well as datasets captured with different sensors. As a result of the above, a method tailored for a specific dataset might fail to reproduce good results for another. Since there is no ground truth information for comparison, the evaluation is usually done through visual inspection of the final result, or in some cases, with the use of colour charts, even though the latter is more commonly used in the cases of image restoration techniques.

In contrast to image enhancement techniques, image restoration recovers a deteriorated image using a model of the degradation and the original image formation. To accomplish that, the methods require various parameters, such as attenuation and diffusion coefficients, as well as the camera to object distance across the whole scene. Additionally, in many of the methods that were presented in Section 2.2, camera-specific information, such as the response function, is essential for image restoration. These techniques produce very realistic results and are able to represent the UW scenery as if it is captured in the air. Although many methods have contributed significantly to the topic of UW colour correction, they are difficult to replicate, either because of the unique scenarios under which the methods were applied or due to the lack of substantial information provided by the contributors. Regarding the former, based on the contributions that have been reviewed, each method was constructed based on specific equipment and scenery as well as the physical properties of water. Due to these factors, trying to replicate any of these methods might lead to suboptimal or false results. Additionally, a major limitation in these methods appears when we deal with scenes that are at depths greater than 20 m UW. In these cases, the presence of natural light is almost non-existent, which means that methods that rely on natural lighting to be present, such as the one described by [17,37,38], become obsolete. In that case, methods such as the ones described in [32,36] are suited to address the issue of UW image colour correction. Although these methods produced very good and realistic results, they are extremely difficult to implement due to the need for very specific and expensive equipment, such as AUVs with fully calibrated cameras and strobe systems that allow researchers to identify information regarding the angle that the strobe lights hit the surface, as well as the distance between the cameras and the strobes. The methods we reviewed suffer from the same issue regarding the evaluation as the methods related to image enhancement, which is the lack of true information regarding the colours of the scene. The evaluation in most cases is done through some comparisons and statistical analysis of the RGB values of colour charts UW in the restored images and in the air. 

As it was explained above, the rise of artificial intelligence methods has contributed, in the last few years, significantly to the topic of UW image colour correction. Many techniques, such as CNN, GAN, MRF and many more, have been developed and utilise large datasets in order to train algorithms that can predict the true colour of the UW scene. As it is explained, such methods require an enormous number of images paired with the appropriate “ground truth” information in order to achieve satisfying predictions. This on its own is considered a major limitation due to the lack of ground truth. Similar to the methods that were showcased in Section 2.3, most contributions either use online datasets that do not carry any information related to the scene geometry, or they utilise synthetic/artificial datasets to achieve the desired results. Even though these methods can provide some relatively realistic results, they cannot be considered superior to image restoration methods due to the lack of water’s physical properties and image formation model as well as the scene geometry. Regarding the evaluation, in the case of AI methods, that is done in its majority through visual inspection, or in specific cases through the effectiveness that they provide in subsequent processes, such as image feature matching.

Based on the reviewed papers, the absence of a common performance indicator is noticeable. Certain papers lack the quantitative evaluation overall, where others address it via various individual metrics for the performance evaluation. Some of the papers that were reviewed, such as [23,43,45], used different individual performance metrics from each other for quantitative evaluation, where others, such as [36,37,38], used metrics derived from statistical analysis between the results and ground truth. Specifically, the lack of universal performance indicators is the source of issues related to the quantitative comparison between the reviewed methods, highlighting, even more, the lack of a benchmark case that can be utilised for these types of comparisons. Furthermore, a common practice for evaluating the effectiveness of the contributions described in this paper is either performing a visual inspection or subsequent procedures. This approach can potentially lead to unreliable outcomes since no independent criterion seems to exist that allows an objective comparison. Recently, many contributions related to UW image colour correction with the use of AI methods [56,57,58,59] try to address this issue by utilising large UW databases, along with their chromatically corrected counterparts. Such a database is the one provided by Li et al. [60].

The best papers based on this current review cannot be identified from quantitative comparisons due to the lack of common metrics as previously stated. However, regarding the impact of visual results as well as the residuals from ground truth data, the best methods that were identified are [36,38]. These two methods utilise the full scene’s 3D geometry as well as the water’s physical properties, and both provide very realistic results on the colour corrected images, each one in different scenarios respectively (deep waters with the use of ROV and artificial light vs. shallow waters with the presence of natural light).

## 4. Conclusions

In this paper, we tried to provide a coherent and unbiased literature review on methodologies engaged with the topic of UW image colour correction from the last 10–15 years, something that has not be done in such an extent since 2010. As it has been showcased in this paper, the advancements in the field of UW image colour correction and restoration have progressed significantly in recent years. Many contributions have produced significant results in terms of the reconstruction of a realistic scene without the presence of water. The methods vary from stochastic based to methods that fully utilise the water’s physical properties and the scene’s geometry. Although certain methods and tools provide results beyond expectations, it could be said that none can be applied universally by researchers and experts. That is due to the unique scenario that each case was evaluated and applied on or due to the very specific and sometimes expensive equipment that was used. Potentially, a case can be made that methods utilising the physical properties of water and scene geometry can be fused with artificial intelligence methods. Since there is a lot of work in AI methods nowadays and a bloom in UW data sets, this approach could be promising. Since the main problem of ML and DL workflows is the existence or not of “ground truth” data, methods that utilise the UW formation model can bypass this issue and provide a “ground truth”. This will potentially allow for a universal approach and solution regarding UW image colour restoration without the use of very specific equipment or without the need of knowing the physical properties of water in any scenario. Thus, it is safe to say that the combination of AI with image restoration techniques that utilise specific or various image formation models might be the best solution. This will also allow scientists in the domains of marine biology, archaeology or even video games, the latter through virtual and augmented reality applications, to have a more approachable way of obtaining a realistic representation of the UW environment. Moreover, this kind of fusion will enable the use and colour restoration of archived UW datasets.

## Figures and Tables

**Figure 1 sensors-21-05690-f001:**
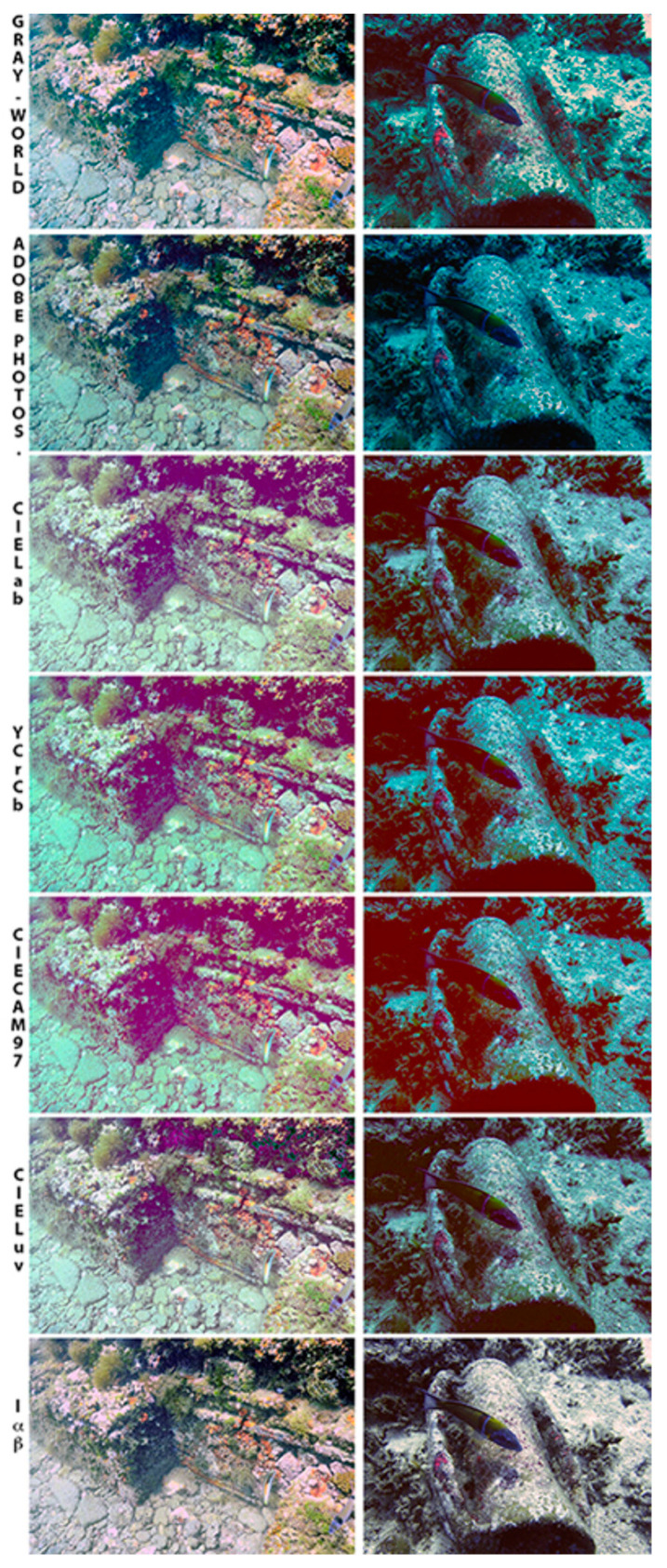
Comparison among different colour correction methods. Image from [24].

**Figure 2 sensors-21-05690-f002:**
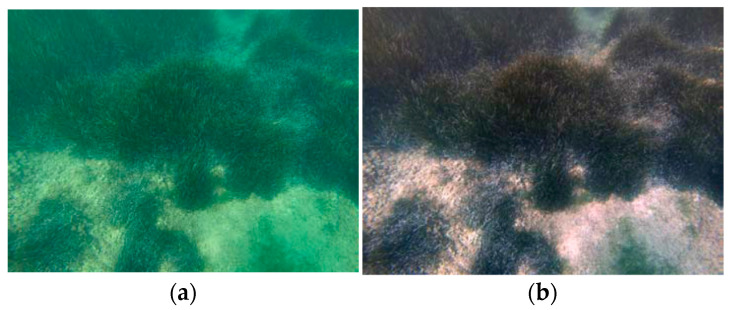
Three real cases of non-uniformly illuminated images processed with the proposed method of Bianco and Neumann. (**a**,**c**,**e**) are the uncorrected images and (**b**,**d**,**f**) are the results after the correction. Images from [27].

**Figure 3 sensors-21-05690-f003:**
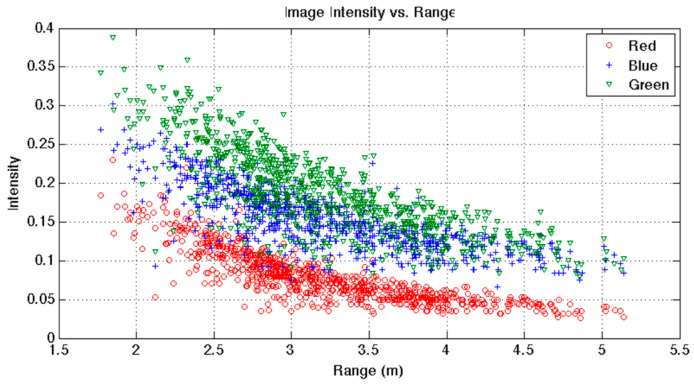
Image pixel intensity decay based on camera-to-object distance from the centre pixel. Image from [32].

**Figure 4 sensors-21-05690-f004:**
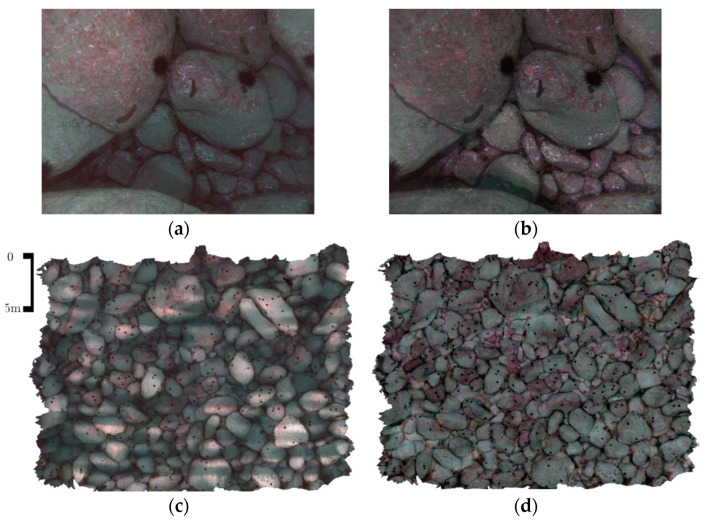
(**a**) Restored image using a non-depth-based approach, (**b**) image with water attenuation compensation, (**c**) 3D texture of the scene using non-depth-based corrected textures and (**d**) 3D texture of the scene with water attenuation corrected textures. Images from [32].

**Figure 5 sensors-21-05690-f005:**
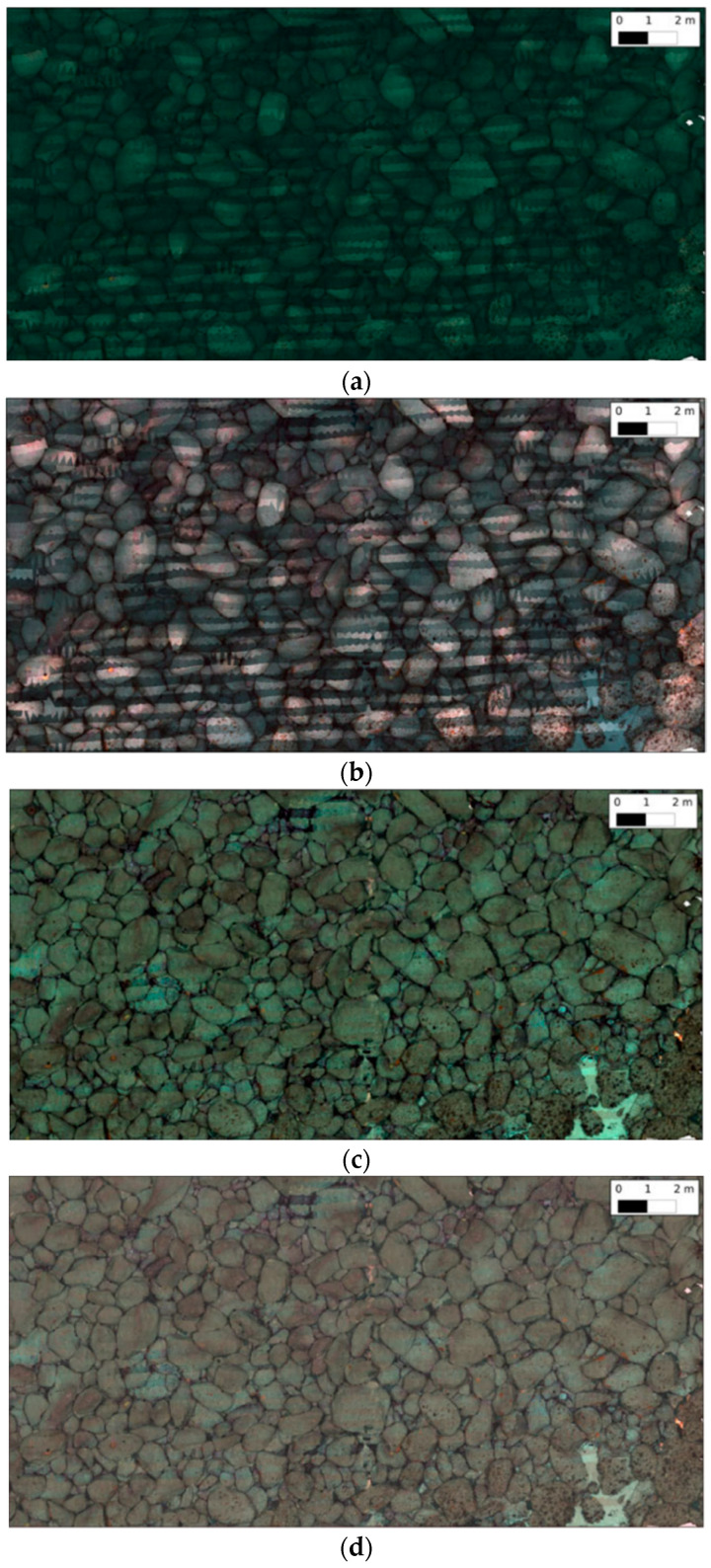
(**a**) Uncorrected UW orthophotomosaic, (**b**) Grey world correction scheme, (**c**) UW image formation model correction scheme and (**d**) UW image formation model correction scheme with camera/strobe spectral processing. Images from [36]. The contents of this figure have been published in “True Color Correction of Autonomous Underwater Vehicle Imagery”, Journal of Field Robotics, Volume 33, Issue 6, published by John Wiley and Sons.

**Figure 6 sensors-21-05690-f006:**
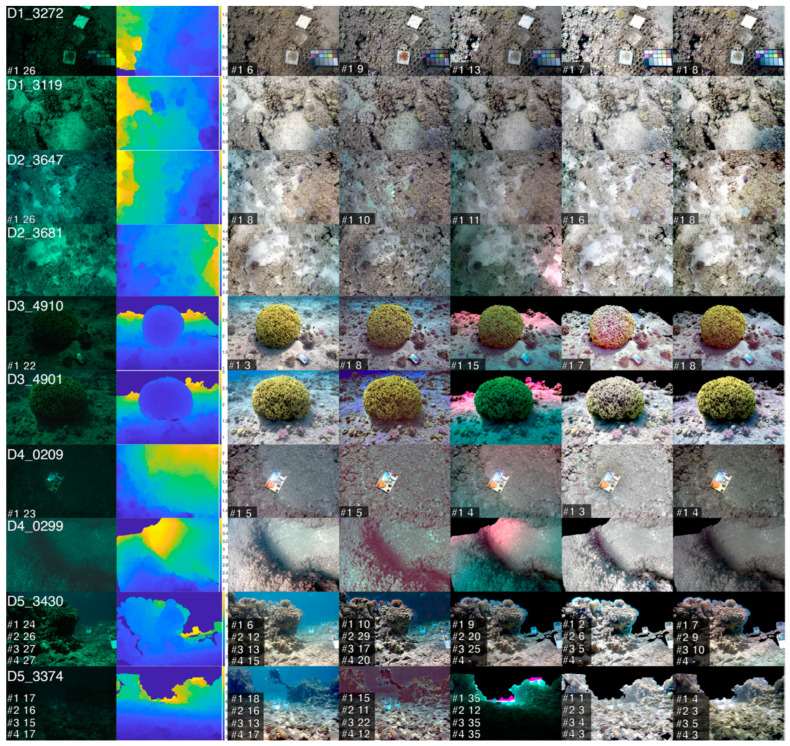
Results presented is Sea-thru. From left to right: raw images, depth maps, S1, S2, S3, S4 and S5 (Sea-thru). Image from [38].

**Figure 7 sensors-21-05690-f007:**
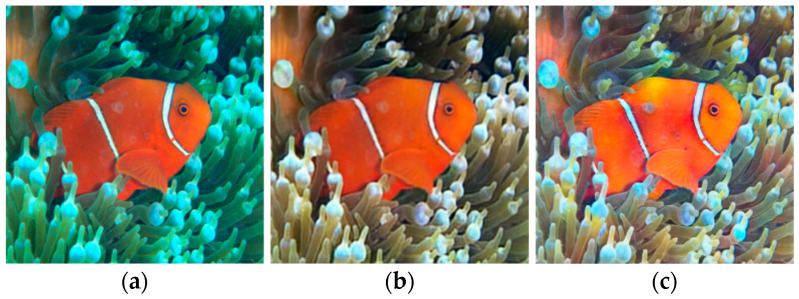
One example presented by Hashisho et al. (**a**) is the input image, (**b**) the result produced by UGAN and (**c**) the result produced by UDAE. Images from [47].

**Table 1 sensors-21-05690-t001:** Brief description of the main methods described in Section 2.

Method	Description	Characteristics and Dataset	Image Quality Evaluation	Quantitative Metrics
**Image Enhancement Methods**				
Ancuti et al. [23]	“A framework based on fusion that combines many filters to enhance UW images based on a single input. A robust white balancing technique specialized for UW scenes”	“White balancing, and fusion of several applications such as image compositing multispectral video enhancement defogging and HDR imaging of UW Degraded Images”	“Comparison with other methods and evaluation based on feature point matching quality”	“Image Quality Metric as proposed by [61]. The metric estimates the Loss of visible contrast, Amplification of invisible features and Contrast reversal”
Bianco et al. [24]	“Colour correction of UW images by using the lαβ colour space”	“Transformation of UW images from the RGB colour space to the lαβ colour space”	“Comparisons with other colour spaces and effectiveness in 3D reconstruction application”	---
Wu et al. [25]	“A systematic approach for UW image restoration based on colour correction and nonlocal prior”	“Eliminate UW image colour distortion to make the corrected UW images look like hazy, and then use a dehazing approach to remove backscattering”	“Visual Inspection and comparison with two other methods”	---
Liu et al. [26]	“Colour correction scheme based on local surface reflectance statistics for UW images”	“Image Segmentation and Illumination estimation on UW degraded images using different image patches”	“Application of the method on images downloaded from the internet and comparison with four other approaches”	---
Bianco et al. [27]	“UW Image colour correction based on the gray-world assumption applied in the Ruderman-lαβ opponent colour space”	“Adaptation of the correction to the nonuniform illumination using a local estimation of the illuminant color by average computation with a moving window around all pixels. This is applied to Images captured in different depths and working distances”	“Comparison of the local approach the authors developed with a global approach through visual inspection”	---
**Image Restoration Methods**				
Bryson et al. [32]	“Exploitation of the 3D structure of the scene generated using structure-from-motion and photogrammetry techniques accounting for distance-based attenuation, vignetting and lighting pattern, and improves the consistency of photo-textured 3D models”	“Colour Correction method applied to UW images captured by an AUV collected in two different UW environments. The colour correction relies to the gray world assumption while considering the scene’s geometry”	“Comparison of image texture standard deviation for the same corrected textures and comparison of 3D photo-textured sceneries utilizing standard and complete water attenuation colour corrected textures”	“Measuring the inconsistency in overlapping images of common Objects”
Galdran et al. [34]	“A Red Channel approach for UW photos is presented, which may be seen as a version of the Dark Channel approach for atmospheric image dehazing”	“Pixels of UW images that lie at the maximum scene depth are picked with respect to the camera. Then the transmission from the Red channel is estimated by extending the Dark Channel prior method”	“Visual Inspection and comparison with five other methods. The comparison is carried out through statistical analysis”	“Calculation of relative dispersion between colour channels and their standard deviation thus obtaining 2 different scores, μ_diff_ and σ_diff_. Additionally the coefficient λ is introduced which is considered as a measure of saturation”
Bryson et al. [36]	“A formation model for calculating the true colour of scenery taken from an UW automated vehicle with strobes is proposed”	“The model takes into account the vignetting, attenuation, backscatter coefficient as well as the system geometry meaning that there is information regarding the angle that the light from the strobes hits the object”	“Visual Inspection and the use of calibrated colour charts”	“Evaluation of normalized image intensities on the corrected images with ground truth values using Macbeth colour chart”
Akkaynak and Treibitz [37]	“Introduction of a revised UW image formation model that takes into account wideband coefficients of backscatter are different than those of direct transmission”	“Application of the model in various UW scenes captured in shallow waters. The images captured are then photogrammetrically processed to acquire the scene geometry. Using the camera to object distance as well as the true RGB values of colour charts in the scene, the coefficients of the revised model are estimated”	“Visual inspection and statistical analysis using ground truth RGB values of colour charts. The evaluation is done for seven scenarios”	“Statistical analysis of errors through the use of colour charts in various scenarios”
Akkaynak and Treibitz [38]	“Application of the revised formation model introduced in [37] with some minor changes creating a pipeline for colour reconstruction”	“Multiple datasets in shallow waters from 5 to 15 m UW are used in order to apply the pipeline of Sea-thru”	“Visual inspection and statistical analysis using ground truth RGB values of colour charts. The evaluation is done for 4 other scenarios that then are compared with the results provided by Sea-thru”	“RGB angular error $\;\bar{\psi }$ between the six grayscale patches of each chart and a pure gray colour, averaged per chart. Lower $\;\bar{\psi }$ value indicates better correction.” “Average errors for the dataset are: raw: 20.57, S1: 12.49, S2: 14.38, S3: 21.77, S4: 4.13, S5: (Sea-thru) 6.33”
**Artificial Intelligence Methods**				
Torres-Méndez et al. [42]	“Colour restoration of AUV acquired images using statistical priors and Markov Random Fields”	“Two scenarios are applied: (1) UW web images “ground truth” and then are distorted for training. (2) acquisition of UW video by AUV with and without strobes”	“Visual Inspection and comparison with ground truth values”	“CIElab Euclidean distance differences is the similarity measure used to select possible candidates to define the compatibility functions and also to evaluate the performance of this method. Additionally, mean absolute residual (MAR) error between the ground truth and the colour corrected images is computed to evaluate the performance of the algorithm” “For 4 different examples the MAR errors are 9.43, 9.65, 9.82, and 12.20, respectively”
Ponce-Hinestroza et al. [43]	“A Spatial Chromatic-MRF Model that accounts for the spatial domain of the images”	“Real UW videos and images are used to apply with Visual Attention White World Assumption (VAWWA) and MRF-BP”	“Visual Inspection and a similarity measure from statistical analysis”	“They define a similarity measure S which is an Euclidean norm between a current vector of statistical parameters and the same vector in the previous frame” “S > 0.01 for a feasible training for the MRF model”
Ponce-Hinestroza [45]	“An automatic way to generate a training set for a MRF designed to recover the colour in a video sequence based on existing colour restoration techniques reported in literature”	“A multiple colour space analysis and processing stage is done to automatically recover the colour in an image frame of an UW video sequence to be used as a training set in the MRF model”	“Visual Inspection and comparison with other known methods from the literature”	“They define a similarity measure S which is an Euclidean norm between a current vector of statistical parameters and the same vector in the previous frame” “S > 0.01 for a feasible training for the MRF model”
Li et al. [46]	“A weakly supervised model for UW image colour correction, which maps the colour from the scenes of UW into the scenes of air without any explicit pair labels”	“The model takes as input an UW image and directly outputs a coloured image as if it was taken without the water”	“Visual Inspection and user study were implemented”	“The user study utilized metric from 1 (worst) to 8 (best). The scores for the method introduced applied in 5 images is as follows: 7.5, 7.5, 6.2, 6.2, 6.6”
Hashisho et al. [47]	“A UDAE model was developed, which is a deep learning network based on a single denoising autoencoder using U-Net as CNN architecture”	“A synthetic dataset is constructed using a generative deep learning method. The dataset has a combination of different UW scenarios”	“Visual Inspection and comparison with UGAN algorithm”	“Three metrics were used for comparison with UGAN. MSE, SSIM and MS-SSIM-L1. MSE and MS-SSIM-L1 give a score 0 for identical images, while SSIM gives a score 1. For UDAE the metric values were 0.0028, 0.9653, 0.0753 where for UGAN were 0.0061, 0.9186, 0.1415”

## Data Availability

Not applicable.

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
