# Peer review of "An Extensive Literature Review on Underwater Image Colour Correction"

_sensors, 2021, doi:10.3390/s21175690_

Round 1

Reviewer 1 Report

The paper “An Extensive Literature Review on Underwater Image Colour Correction” is interesting, the results of the reviewed papers are presented in a complete and logical manner, and a description of their strengths and weaknesses allows the reader to draw conclusions about the applicability of the methods. The comparison table at the end is an important and handy tool.

My main question relates to the choice of the image quality evaluation criteria. As can be seen from the works presented in the review, in most cases it is subjective and based on visual inspection. I would like to hear the opinion of the authors of the review about the possibility of choosing an independent criterion that allows an objective comparison of the methods under consideration.

Page 4, line 139. Perhaps it should be clarified that the conclusion that the object looks gray refers to the second column of images in Fig. 2.

Fig. 3. Depth images. It is not clear from the explanations of the authors how they are generated.

Fig. 6. At what optical parameters of water was this drawing obtained?

Page 11 line 276. "Very good methodology" does not sound specific, it is better to use another expression.

Page 13 line 345. What is the difference between the five scenarios considered in [38]?

As for section 2.3 Artificial intelligence methods, I agree with the authors of the review about the prospects of such methods, although the works reviewed can be assessed only as a proof of concept, problems with neural networks generalization are visible.

Author Response

The answers to the questions are all inside the pdf file i uploaded

Reviewer 2 Report

This review presents and discusses recent scientific literature in the domain of Under Water ImageColourEnhancement and Restoration. The topic is exotic but interesting and the review can offer an overview even for other applications.

The methodological aspects related to the selection of the papers must be included in the documents. I strongly suggest following the guidelines provided by PRISMA in their checklist http://www.prisma-statement.org/documents/PRISMA_2020_checklist.docx. Furthermore, the PRISMA flow diagram must be included http://www.prisma-statement.org/PRISMAStatement/FlowDiagram.

Concerning the results described in the review, I think that most of them are too qualitative. It's ok to show images from other papers but the authors should make an effort to provide some quantitative results about the effectiveness of the methods focusing on some relevant metrics. This can be also be included in a summary table.

Author Response

The answers to the reviewer's comments are in the pdf file attached below. Additionally, we need to clarify to the reviewer that due to curtain issues with copyrights, many of the figures included in the manuscript do not exist anymore in the revised version.

Round 2

Reviewer 2 Report

The authors did not fully addressed my concerns neither on the methodological aspects (How many papers were screened? How the selection fo the papers was performed  (e.g. inclusion/exclusion criteria) ?) nor on the quantitative results for comparison of the methods proposed in the review.

Which are the best methods for image colour enhancement/restoration? What are their performances? Is it possible to compare their performances based on quantitative metrics?

Author Response

Comments addressed in the attached pdf
